# Compressive Feature Learning

**Hristo S. Paskov**
Department of Computer Science
Stanford University
hpaskov@cs.stanford.edu

**Robert West**
Department of Computer Science
Stanford University
west@cs.stanford.edu

**John C. Mitchell**
Department of Computer Science
Stanford University
mitchell@cs.stanford.edu

**Trevor J. Hastie**
Department of Statistics
Stanford University
hastie@stanford.edu

## Abstract

This paper addresses the problem of unsupervised feature learning for text data. Our method is grounded in the principle of minimum description length and uses a dictionary-based compression scheme to extract a succinct feature set. Specifically, our method finds a set of word $k$-grams that minimizes the cost of reconstructing the text losslessly. We formulate document compression as a binary optimization task and show how to solve it approximately via a sequence of reweighted linear programs that are efficient to solve and parallelizable. As our method is unsupervised, features may be extracted once and subsequently used in a variety of tasks. We demonstrate the performance of these features over a range of scenarios including unsupervised exploratory analysis and supervised text categorization. Our compressed feature space is two orders of magnitude smaller than the full $k$-gram space and matches the text categorization accuracy achieved in the full feature space. This dimensionality reduction not only results in faster training times, but it can also help elucidate structure in unsupervised learning tasks and reduce the amount of training data necessary for supervised learning.

## 1 Introduction

Machine learning algorithms rely critically on the features used to represent data; the feature set provides the primary interface through which an algorithm can reason about the data at hand. A typical pitfall for many learning problems is that there are too many potential features to choose from. Intelligent subselection is essential in these scenarios because it can discard noise from irrelevant features, thereby requiring fewer training examples and preventing overfitting. Computationally, a smaller feature set is almost always advantageous as it requires less time and space to train the algorithm and make inferences [10, 9].

Various heuristics have been proposed for feature selection, one class of which work by evaluating each feature separately with respect to its discriminative power. Some examples are document frequency, chi-square value, information gain, and mutual information [26, 9]. More sophisticated methods attempt to achieve feature sparsity by optimizing objective functions containing an $L_1$ regularization penalty [25, 27].

Unsupervised feature selection methods [19, 18, 29, 13] are particularly attractive. First, they do not require labeled examples, which are often expensive to obtain (e.g., when humans have to provide them) or might not be available in advance (e.g., in text classification, the topic to be retrieved might be defined only at some later point). Second, they can be run a single time in an offline preprocessing

step, producing a reduced feature space that allows for subsequent rapid experimentation. Finally, a good data representation obtained in an unsupervised way captures inherent structure and can be used in a variety of machine learning tasks such as clustering, classification, or ranking.

In this work we present a novel unsupervised method for feature selection for text data based on ideas from data compression and formulated as an optimization problem. As the universe of potential features, we consider the set of all word $k$-grams.[1] The basic intuition is that substrings appearing frequently in a corpus represent a recurring theme in some of the documents, and hence pertain to class representation. However, it is not immediately clear how to implement this intuition. For instance, consider a corpus of NIPS papers. The bigram 'supervised learning' will appear often, but so will the constituent unigrams 'supervised' and 'learning'. So shall we use the bigram, the two separate unigrams, or a combination, as features?

Our solution invokes the principle of minimum description length (MDL) [23]: First, we compress the corpus using a dictionary-based lossless compression method. Then, the substrings that are used to reconstruct each document serve as the feature set. We formulate the compression task as a numerical optimization problem. The problem is non-convex, but we develop an efficient approximate algorithm that is linear in the number of words in the corpus and highly parallelizable. In the example, the bigram 'supervised learning' would appear often enough to be added to the dictionary; 'supervised' and 'learning' would also be chosen as features if they appear separately in combinations other than 'supervised learning' (because the compression paradigm we choose is lossless).

We apply our method to two datasets and compare it to a canonical bag-of-$k$-grams representation. Our method reduces the feature set size by two orders of magnitude without incurring a loss of performance on several text categorization tasks. Moreover, it expedites training times and requires significantly less labeled training data on some text categorization tasks.

## 2   Compression and Machine Learning

Our work draws on a deep connection between data compression and machine learning, exemplified early on by the celebrated MDL principle [23]. More recently, researchers have experimented with off-the-shelf compression algorithms as machine learning subroutines. Instances are Frank et al.'s [7] compression-based approach to text categorization, as well as compression-based distance measures, where the basic intuition is that, if two texts $x$ and $y$ are very similar, then the compressed version of their concatenation $xy$ should not be much longer than the compressed version of either $x$ or $y$ separately. Such approaches have been shown to work well on a variety of tasks such as language clustering [1], authorship attribution [1], time-series clustering [6, 11], anomaly detection [11], and spam filtering [3].

Distance-based approaches are akin to kernel methods, and thus suffer from the problem that constructing the full kernel matrix for large datasets might be infeasible. Furthermore, Frank et al. [7] deplore that "it is hard to see how efficient feature selection could be incorporated" into the compression algorithm. But Sculley and Brodley [24] show that many compression-based distance measures can be interpreted as operating in an implicit high-dimensional feature space, spanned by the dictionary elements found during compression. We build on this observation to address Frank et al.'s above-cited concern about the impossibility of feature selection for compression-based methods. Instead of using an off-the-shelf compression algorithm as a black-box kernel operating in an implicit high-dimensional feature space, we develop an optimization-based compression scheme *whose explicit job it is* to perform feature selection.

It is illuminating to discuss a related approach suggested (as future work) by Sculley and Brodley [24], namely "to store substrings found by Lempel–Ziv schemes as explicit features". This simplistic approach suffers from a serious flaw that our method overcomes. Imagine we want to extract features from an entire corpus. We would proceed by concatenating all documents in the corpus into a single large document $D$, which we would compress using a Lempel–Ziv algorithm. The problem is that the extracted substrings are dependent on the order in which we concatenate the documents to form the input $D$. For the sake of concreteness, consider LZ77 [28], a prominent member of the Lempel–Ziv family (but the argument applies equally to most standard compression algorithms). Starting from the current cursor position, LZ77 scans $D$ from left to right, consuming characters until it

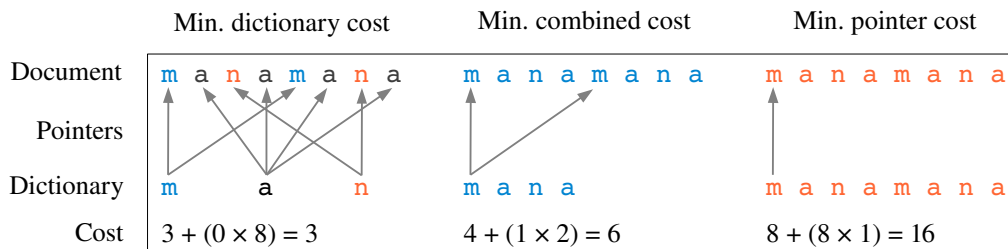

| Min. dictionary cost | Min. combined cost | Min. pointer cost |
|---|---|---|
| Document: m a n a m a n a | m a n a m a n a | m a n a m a n a |
| Pointers | | |
| Dictionary: m a n | m a n a | m a n a m a n a |
| Cost: $3 + (0 \times 8) = 3$ | $4 + (1 \times 2) = 6$ | $8 + (8 \times 1) = 16$ |

Figure 1: Toy example of our optimization problem for text compression. Three different solutions shown for representing the 8-word document $D = manamana$ in terms of dictionary and pointers. Dictionary cost: number of characters in dictionary. Pointer cost: $\lambda \times$ number of pointers. Costs given as dictionary cost + pointer cost. **Left:** dictionary cost only ($\lambda = 0$). **Right:** expensive pointer cost ($\lambda = 8$). **Center:** balance of dictionary and pointer costs ($\lambda = 1$).

has found the longest prefix matching a previously seen substring. It then outputs a pointer to that previous instance—we interpret this substring as a feature—and continues with the remaining input string (if no prefix matches, the single next character is output). This approach produces different feature sets depending on the order in which documents are concatenated. Even in small instances such as the 3-document collection $\{D_1 = abcd, D_2 = ceab, D_3 = bce\}$, the order $(D_1, D_2, D_3)$ yields the feature set $\{ab, bc\}$, whereas $(D_2, D_3, D_1)$ results in $\{ce, ab\}$ (plus, trivially, the set of all single characters).

As we will demonstrate in our experiments section, this instability has a real impact on performance and is therefore undesirable. Our approach, like LZ77, seeks common substrings. However, our formulation is not affected by the concatenation order of corpus documents and does not suffer from LZ77's instability issues.

## 3   Compressive Feature Learning

The MDL principle implies that a good feature representation for a document $D = x_1 x_2 \ldots x_n$ of $n$ words minimizes some description length of $D$. Our dictionary-based compression scheme accomplishes this by representing $D$ as a *dictionary*—a subset of $D$'s substrings—and a sequence of *pointers* indicating where copies of each dictionary element should be placed in order to fully reconstruct the document. The compressed representation is chosen so as to minimize the cost of storing each dictionary element in plaintext as well as all necessary pointers. This scheme achieves a shorter description length whenever it can reuse dictionary elements at different locations in $D$.

For a concrete example, see Fig. 1, which shows three ways of representing a document $D$ in terms of a dictionary and pointers. These representations are obtained by using the same pointer storage cost $\lambda$ for each pointer and varying $\lambda$. The two extreme solutions focus on minimizing either the dictionary cost ($\lambda = 0$) or the pointer cost ($\lambda = 8$) solely, while the middle solution ($\lambda = 1$) trades off between minimizing a combination of the two. We are particularly interested in this tradeoff: when all pointers have the same cost, the dictionary and pointer costs pull the solution in opposite directions. Varying $\lambda$ allows us to 'interpolate' between the two extremes of minimum dictionary cost and minimum pointer cost. In other words, $\lambda$ can be interpreted as tracing out a regularization path that allows a more flexible representation of $D$.

To formalize our compression criterion, let $\mathcal{S} = \{x_i \ldots x_{i+t-1} \mid 1 \le t \le k, 1 \le i \le n-t+1\}$ be the set of all unique $k$-grams in $D$, and $\mathcal{P} = \{(s, l) \mid s = x_l \ldots x_{l+|s|-1}\}$ be the set of all $m = |\mathcal{P}|$ (potential) pointers. Without loss of generality, we assume that $\mathcal{P}$ is an ordered set, i.e., each $i \in \{1, \ldots, m\}$ corresponds to a unique $p_i \in \mathcal{P}$, and we define $J(s) \subset \{1, \ldots, m\}$ to be the set of indices of all pointers which share the same string $s$. Given a binary vector $w \in \{0, 1\}^m$, $w$ *reconstructs* word $x_j$ if for some $w_i = 1$ the corresponding pointer $p_i = (s, l)$ satisfies $l \le j < l + |s|$. This notation uses $w_i$ to indicate whether pointer $p_i$ should be used to reconstruct (part of) $D$ by pasting a copy of string $s$ into location $l$. Finally, $w$ reconstructs $D$ if every $x_j$ is reconstructed by $w$.

Compressing $D$ can be cast as a binary linear minimization problem over $w$; this bit vector tells us which pointers to use in the compressed representation of $D$ and it implicitly defines the dictionary (a subset of $\mathcal{S}$). In order to ensure that $w$ reconstructs $D$, we require that $Xw \geq \mathbf{1}$. Here $X \in \{0,1\}^{n \times m}$ indicates which words each $w_i = 1$ can reconstruct: the $i$-th column of $X$ is zero everywhere except for a contiguous sequence of ones corresponding to the words which $w_i = 1$ reconstructs. Next, we assume the pointer storage cost of setting $w_i = 1$ is given by $d_i \geq 0$ and that the cost of storing any $s \in \mathcal{S}$ is $c(s)$. Note that $s$ must be stored in the dictionary if $\|w_{J(s)}\|_\infty = 1$, i.e., some pointer using $s$ is used in the compression of $D$. Putting everything together, our lossless compression criterion is

$$\underset{w}{\text{minimize}} \quad w^T d + \sum_{s \in \mathcal{S}} c(s) \|w_{J(s)}\|_\infty \qquad \text{subject to} \quad Xw \geq \mathbf{1}, \quad w \in \{0,1\}^m. \qquad (1)$$

Finally, multiple documents can be compressed jointly by concatenating them in any order into a large document and disallowing any pointers that span document boundaries. Since this objective is invariant to the document concatenating order, it does not suffer from the same problems as LZ77 (cf. Section 2).

## 4   Optimization Algorithm

The binary constraint makes the problem in (1) non-convex. We solve it approximately via a series of related convex problems $P^{(1)}, P^{(2)}, \ldots$ that converge to a good optimum. Each $P^{(i)}$ relaxes the binary constraint to only require $\mathbf{0} \leq w \leq \mathbf{1}$ and solves a weighted optimization problem

$$\underset{w}{\text{minimize}} \quad w^T \tilde{d}^{(i)} + \sum_{s \in \mathcal{S}} c(s) \|D^{(i)}_{J(s)J(s)} w_{J(s)}\|_\infty \qquad \text{subject to} \quad Xw \geq \mathbf{1}, \quad \mathbf{0} \leq w \leq \mathbf{1}. \qquad (2)$$

Here, $D^{(i)}$ is an $m \times m$ diagonal matrix of positive weights and $\tilde{d}^{(i)} = D^{(i)} d$ for brevity. We use an iterative reweighting scheme that uses $D^{(1)} = I$ and $D^{(i+1)}_{jj} = \max\left\{1, (w^{(i)}_j + \epsilon)^{-1}\right\}$, where $w^{(i)}$ is the solution to $P^{(i)}$. This scheme is inspired by the iterative reweighting method of Candès et al. [5] for solving problems involving $L_0$ regularization. At a high level, reweighting can be motivated by noting that (2) recovers the correct *binary* solution if $\epsilon$ is sufficiently small and we use as weights a nearly binary solution to (1). Since we do not know the correct weights, we estimate them from our best guess to the solution of (1). In turn, $D^{(i+1)}$ punishes coefficients that were small in $w^{(i)}$ and, taken together with the constraint $Xw \geq \mathbf{1}$, pushes the solution to be binary.

**ADMM Solution**   We demonstrate an efficient and parallel algorithm to solve (2) based on the Alternating Directions Method of Multipliers (ADMM) [2]. Problem (2) is a linear program solvable by a general purpose method in $O(m^3)$ time. However, if all potential dictionary elements are no longer than $k$ words in length, we can use problem structure to achieve a run time of $O(k^2 n)$ per step of ADMM, i.e., linear in the document length. This is helpful because $k$ is relatively small in most scenarios: long $k$-grams tend to appear only once and are not helpful for compression. Moreover, they are rarely used in NLP applications since the relevant signal is captured by smaller fragments.

ADMM is an optimization framework that operates by splitting a problem into two subproblems that are individually easier to solve. It alternates solving the subproblems until they both agree on the solution, at which point the full optimization problem has been solved. More formally, the optimum of a convex function $h(w) = f(w) + g(w)$ can be found by minimizing $f(w) + g(z)$ subject to the constraint that $w = z$. ADMM acccomplishes this by operating on the *augmented Lagrangian*

$$\mathcal{L}_\rho(w,z,y) = f(w) + g(z) + y^T(w-z) + \frac{\rho}{2}\|w-z\|_2^2. \qquad (3)$$

It minimizes $\mathcal{L}_\rho$ with respect to $w$ and $z$ while maximizing with respect to dual variable $y \in \mathbb{R}^m$ in order to enforce the condition $w = z$. This minimization is accomplished by, at each step, solving for $w$, then $z$, then updating $y$ according to [2]. These steps are repeated until convergence.

Dropping the $D^{(i)}$ superscripts for legibility, we can exploit problem structure by splitting (2) into

$$f(w) = w^T \tilde{d} + \sum_{s \in \mathcal{S}} c(s) \|D_{J(s)J(s)} w_{J(s)}\|_\infty + I_+(w), \qquad g(z) = I_+(Xz - \mathbf{1}) \qquad (4)$$

where $I_+(\cdot)$ is 0 if its argument is non-negative and $\infty$ otherwise. We eliminated the $w \le \mathbf{1}$ constraint because it is unnecessary—any optimal solution will automatically satisfy it.

**Minimizing $w$**  The dual of this problem is a quadratic knapsack problem solvable in linear expected time [4], we provide a similar algorithm that solves the primal formulation. We solve for each $w_{J(s)}$ separately since the optimization is separable in each block of variables. It can be shown [21] that $w_{J(s)} = 0$ if $\|D_{J(s)J(s)}^{-1} q_{J(s)}\|_1 \le c(s)$, where $q_{J(s)} = \max\{\rho z_{J(s)} - \tilde{d}_{J(s)} - y_{J(s)}, 0\}$ and the max operation is applied elementwise. Otherwise, $w_{J(s)}$ is non-zero and the $L_\infty$ norm only affects the maximal coordinates of $D_{J(s)J(s)} w_{J(s)}$. For simplicity of exposition, we assume that the coefficients of $w_{J(s)}$ are sorted in decreasing order according to $D_{J(s)J(s)} q_{J(s)}$, i.e., $[D_{J(s)J(s)} q_{J(s)}]_j \ge [D_{J(s)J(s)} q_{J(s)}]_{j+1}$. This is always possible by permuting coordinates. We show in [21] that, if $D_{J(s)J(s)} w_{J(s)}$ has $r$ maximal coordinates, then

$$w_{J(s)_j} = D_{J(s)_j J(s)_j}^{-1} \min\left\{ D_{J(s)_j J(s)_j} q_{J(s)_j}, \frac{\sum_{v=1}^r D_{J(s)_v J(s)_v}^{-1} q_{J(s)_v} - c(s)}{\sum_{v=1}^r D_{J(s)_v J(s)_v}^{-2}} \right\}. \qquad (5)$$

We can find $r$ by searching for the smallest value of $r$ for which exactly $r$ coefficients in $D_{J(s)J(s)} w_J(s)$ are maximal when determined by the formula above. As discussed in [21], an algorithm similar to the linear-time median-finding algorithm can be used to determine $w_{J(s)}$ in linear expected time.

**Minimizing $z$**  Solving for $z$ is tantamount to projecting a weighted combination of $w$ and $y$ onto the polyhderon given by $Xz \ge \mathbf{1}$ and is best solved by taking the dual. It can be shown [21] that the dual optimization problem is

$$\underset{\alpha}{\text{minimize}} \quad \frac{1}{2} \alpha^T H \alpha - \alpha^T (\rho \mathbf{1} - X(y + \rho w)) \qquad \text{subject to} \quad \alpha \ge \mathbf{0} \qquad (6)$$

where $\alpha \in \mathbb{R}_+^n$ is a dual variable enforcing $Xz \ge \mathbf{1}$ and $H = XX^T$. Strong duality obtains and $z$ can be recovered via $z = \rho^{-1}(y + \rho w + X^T \alpha)$.

The matrix $H$ has special structure when $\mathcal{S}$ is a set of $k$-grams no longer than $k$ words. In this case, [21] shows that $H$ is a $(k-1)$–banded positive definite matrix so we can find its Cholesky decomposition in $O(k^2 n)$. We then use an active-set Newton method [12] to solve (6) quickly in approximately 5 Cholesky decompositions. A second important property of $H$ is that, if $N$ documents $n_1, \ldots, n_N$ words long are compressed jointly and no $k$-gram spans two documents, then $H$ is block-diagonal with block $i$ an $n_i \times n_i$ $(k-1)$–banded matrix. This allows us to solve (6) *separately* for each document. Since the majority of the time is spent solving for $z$, this property allows us to parallelize the algorithm and speed it up considerably.

## 5  Experiments

**20 Newsgroups Dataset**  The majority of our experiments are performed on the 20 Newsgroups dataset [15, 22], a collection of about 19K messages approximately evenly split among 20 different newsgroups. Since each newsgroup discusses a different topic, some more closely related than others, we investigate our compressed features' ability to elucidate class structure in supervised and unsupervised learning scenarios. We use the "by-date" 60%/40% training/testing split described in [22] for all classification tasks. This split makes our results comparable to the existing literature and makes the task more difficult by removing correlations from messages that are responses to one another.

**Feature Extraction and Training** We compute a bag-of-$k$-grams representation from a compressed document by counting the number of pointers that use each substring in the compressed version of the document. This method retrieves the canonical bag-of-$k$-grams representation when all pointers are used, i.e., $w = \mathbf{1}$. Our compression criterion therefore leads to a less redundant representation. Note that we extract features for a document corpus by compressing all of its documents jointly and then splitting into testing and training sets. Since this process involves no label information, it ensures that our estimate of testing error is unbiased.

All experiments were limited to using 5-grams as features, i.e., $k = 5$ for our compression algorithm. Each substring's dictionary cost was its word length and the pointer cost was uniformly set to $0 \leq \lambda \leq 5$. We found that an overly large $\lambda$ hurts accuracy more than an overly small value since the former produces long, infrequent substrings, while the latter tends to a unigram representation. It is also worthwhile to note that the storage cost (i.e., the value of the objective function) of the binary solution was never more than 1.006 times the storage cost of the relaxed solution, indicating that we consistently found a good local optimum.

Finally, all classification tasks use an Elastic-Net–regularized logistic regression classifier implemented by `glmnet` [8]. Since this regularizer is a mix of $L_1$ and $L_2$ penalties, it is useful for feature selection but can also be used as a simple $L_2$ ridge penalty. Before training, we normalize each document by its $L_1$ norm and then normalize features by their standard deviation. We use this scheme so as to prevent overly long documents from dominating the feature normalization.

**LZ77 Comparison** Our first experiment demonstrates LZ77's sensitivity to document ordering on a simple binary classification task of predicting whether a document is from the alt.atheism (A) or comp.graphics (G) newsgroup. Features were computed by concatenating documents in different orders: (1) by class, i.e., all documents in A before those in G, or G before A; (2) randomly; (3) by alternating the class every other document. Fig. 5 shows the testing error compared to features computed from our criterion. Error bars were estimated by bootstrapping the testing set 100 times, and all regularization parameters were chosen to minimize testing error while $\lambda$ was fixed at 0.03. As predicted in Section 2, document ordering has a marked impact on performance, with the by-class and random orders performing significantly worse than the alternating ordering. Moreover, order invariance and the ability to tune the pointer cost lets our criterion select a better set of 5-grams.

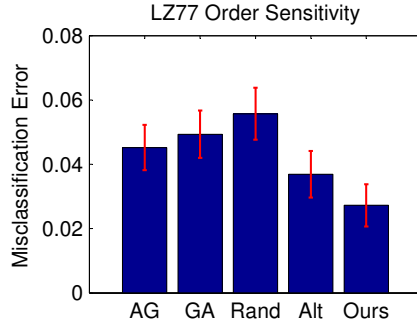

Figure 2: Misclassification error and standard error bars when classifying alt.atheism (A) vs. comp.graphics (G) from 20 Newsgroups. The four leftmost results are on features from running LZ77 on documents ordered by class (AG, GA), randomly (Rand), or by alternating classes (Alt); the rightmost is on our compressed features.

**PCA** Next, we investigate our features in a typical exploratory analysis scenario: a researcher looking for interesting structure by plotting all pairs of the top 10 principal components of the data. In particular, we verify PCA's ability to recover binary class structure for the A and G newsgroups, as well as multiclass structure for the A, comp.sys.ibm.pc.hardware (PC), rec.motorcycles (M), sci.space (S), and talk.politics.mideast (PM) newsgroups. Fig. 3 plots the pair of principal components that best exemplifies class structure using (1) compressed features and (2) all 5-grams. For the sake of fairness, the components were picked by training a logistic regression on every pair of the top 10 principal components and selecting the pair with the lowest training error. In both the binary and multiclass scenarios, PCA is inundated by millions of features when using all 5-grams and cannot display good class structure. In contrast, compression reduces the feature set to tens of thousands (by two orders of magnitude) and clearly shows class structure. The star pattern of the five classes stands out even when class labels are hidden.

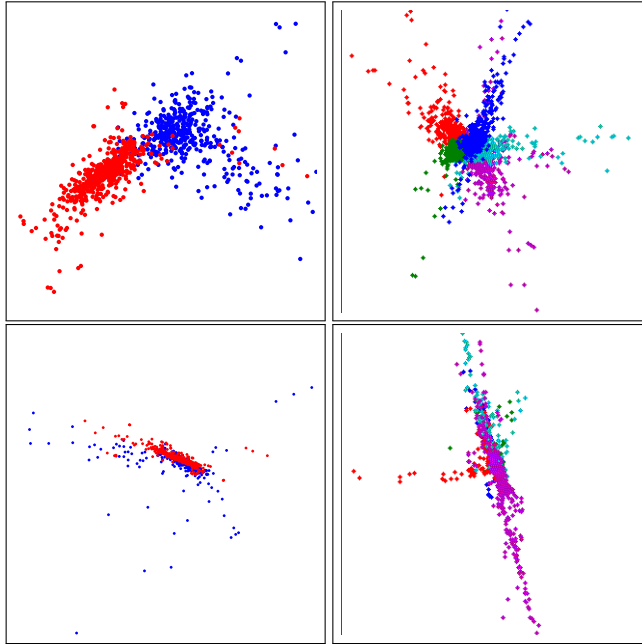

Figure 3: PCA plots for 20 Newsgroups. **Left:** alt.atheism (blue), comp.graphics (red). **Right:** alt.atheism (blue), comp.sys.ibm.pc.hardware (green), rec.motorcycles (red), sci.space (cyan), talk.politics.mideast (magenta). **Top:** compressed features (our method). **Bottom:** all 5-grams.

Table 1: Classification accuracy on the 20 Newsgroups and IMDb datasets

| Method | 20 Newsgroups | IMDb |
|---|---|---|
| Discriminative RBM [16] | 76.2 | — |
| Bag-of-Words SVM [14, 20] | 80.8 | 88.2 |
| Naïve Bayes [17] | 81.8 | — |
| Word Vectors [20] | — | 88.9 |
| **All 5-grams** | 82.8 | 90.6 |
| **Compressed (our method)** | 83.0 | 90.4 |

**Classification Tasks**    Table 1 compares the performance of compressed features with all 5-grams on two tasks: (1) categorizing posts from the 20 Newsgroups corpus into one of 20 classes; (2) categorizing movie reviews collected from IMDb [20] into one of two classes (there are 25,000 training and 25,000 testing examples evenly split between the classes). For completeness, we include comparisons with previous work for 20 Newsgroups [16, 14, 17] and IMDb [20]. All regularization parameters, including $\lambda$, were chosen through 10-fold cross validation on the training set. We also did not $L_1$-normalize documents in the binary task because it was found to be counterproductive on the training set.

Our classification performance is state of the art in both tasks, with the compressed and all-5-gram features tied in performance. Since both datasets feature copious amounts of labeled data, we expect the 5-gram features to do well because of the power of the Elastic-Net regularizer. What is remarkable is that the compression retains useful features without using any label information. There are tens of millions of 5-grams, but compression reduces them to hundreds of thousands (by two orders of magnitude). This has a particularly noticeable impact on training time for the 20 Newsgroups dataset. Cross-validation takes 1 hour with compressed features and 8–16 hours for all 5-grams on our reference computer depending on the sparsity of the resulting classifier.

**Training-Set Size**    Our final experiment explores the impact of training-set size on binary-classification accuracy for the A vs. G and rec.sport.baseball (B) vs. rec.sport.hockey (H) newsgroups. Fig. 4 plots testing error as the amount of training data varies, comparing compressed features to full

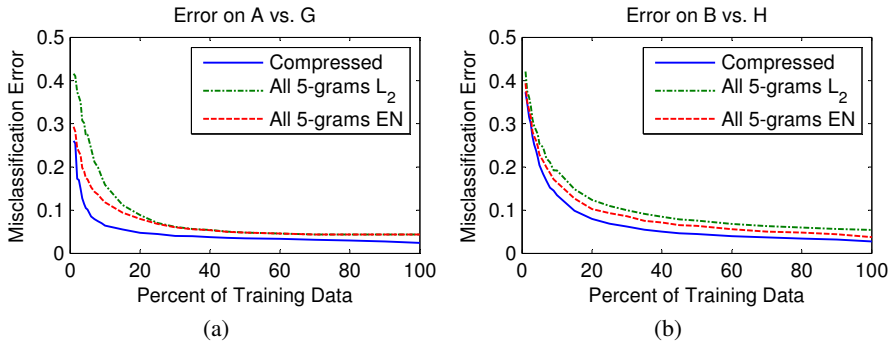

Figure 4: Classification accuracy as the training set size varies for two classification tasks from 20 Newsgroups: **(a)** alt.atheism (A) vs. comp.graphics (G); **(b)** rec.sport.baseball (B) vs. rec.sport.hockey (H). To demonstrate the effects of feature selection, $L_2$ indicates $L_2$-regularization while EN indicates elastic-net regularization.

5-grams; we explore the latter with and without feature selection enabled (i.e., Elastic Net vs. $L_2$ regularizer). We resampled the training set 100 times for each training-set size and report the average accuracy. All regularization parameters were chosen to minimize the testing error (so as to eliminate effects from imperfect tuning) and $\lambda = 0.03$ in both tasks. For the A–G task, the compressed features require substantially less data than the full 5-grams to come close to their best testing error. The B–H task is harder and all three classifiers benefit from more training data, although the gap between compressed features and all 5-grams is widest when less than half of the training data is available. In all cases, the compressed features outperform the full 5-grams, indicating that that latter may benefit from even more training data. In future work it will be interesting to investigate the efficacy of compressed features on more intelligent sampling schemes such as active learning.

## 6 Discussion

We develop a feature selection method for text based on lossless data compression. It is unsupervised and can thus be run as a task-independent, one-off preprocessing step on a corpus. Our method achieves state-of-the-art classification accuracy on two benchmark datasets despite selecting features without any knowledge of the class labels. In experiments comparing it to a full 5-gram model, our method reduces the feature-set size by two orders of magnitude and requires only a fraction of the time to train a classifier. It selects a compact feature set that can require significantly less training data and reveals unsupervised problem structure (e.g., when using PCA).

Our compression scheme is more robust and less arbitrary compared to a setup which uses off-the-shelf compression algorithms to extract features from a document corpus. At the same time, our method has increased flexibility since the target $k$-gram length is a tunable parameter. Importantly, the algorithm we present is based on iterative reweighting and ADMM and is fast enough—linear in the input size when $k$ is fixed, and highly parallelizable—to allow for computing a regularization path of features by varying the pointer cost. Thus, we may adapt the compression to the data at hand and select features that better elucidate its structure.

Finally, even though we focus on text data in this paper, our method is applicable to any sequential data where the sequence elements are drawn from a finite set (such as the universe of words in the case of text data). In future work we plan to compress click stream data from users browsing the Web. We also plan to experiment with approximate text representations obtained by making our criterion lossy.

### Acknowledgments

We would like to thank Andrej Krevl, Jure Leskovec, and Julian McAuley for their thoughtful discussions and help with our paper.

## Footnotes

[1]In the remainder of this paper, the term '$k$-grams' includes sequences of *up to* (rather than exactly) $k$ words.

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
