[Supplementary Material · appendix.pdf]

# Supplementary Material for Compressive Feature Learning

**Hristo S. Paskov**
Department of Computer Science
Stanford University
hpaskov@cs.stanford.edu

**Robert West**
Department of Computer Science
Stanford University
west@cs.stanford.edu

**John C. Mitchell**
Department of Computer Science
Stanford University
mitchell@cs.stanford.edu

**Trevor J. Hastie**
Department of Statistics
Stanford University
hastie@stanford.edu

## 1 Solving with ADMM

We wish to minimize

$$
\begin{aligned}
\underset{w}{\text{minimize}} \quad & d^T w + \sum_{s \in \mathcal{S}} c(s) \| D_{J(s)} w_{J(s)} \|_\infty \\
\text{subject to} \quad & Xw \geq \mathbf{1}, \quad w \geq \mathbf{0}
\end{aligned}
\tag{1}
$$

For the sake of brevity, we take $D_{J(s)}$ to mean $D_{J(s)J(s)}$. With this in mind, the augmented Lagrangian of (1) is

$$
d^T w + \sum_{s \in \mathcal{S}} c(s) \| D_{J(s)} w_{J(s)} \|_\infty + I_+(w) + I_+(Xz - \mathbf{1}) + y^T(w - z) + \frac{\rho}{2} \| w - z \|_2^2
\tag{2}
$$

### 1.1 Solving for $w$

The relevant parts of (2) for $w$ are

$$
d^T w + \sum_{s \in \mathcal{S}} c(s) \| D_{J(s)} w_{J(s)} \|_\infty + I_+(w) + y^T w + \frac{\rho}{2} \| w - z \|_2^2
\tag{3}
$$

Notice that this separates out with respect to $w_{J(s)}$ so we can focus on each group separately. We therefore drop subscripts and use $w$ to refer to $w_{J(s)}$ and $D$ to $D_{J(s)}$. The problem can be restated as

$$
\begin{aligned}
\underset{w}{\text{minimize}} \quad & d^T w + ct + y^T w + \frac{\rho}{2} \| w - z \|_2^2 \\
\text{subject to} \quad & w \geq \mathbf{0}, \quad Dw \leq t\mathbf{1}
\end{aligned}
\tag{4}
$$

We have replaced the $L_\infty$-norm via an epigraph variable transform. Note that we don't need a $t \geq 0$ constraint because it is implied by the existing ones. The Lagrangian is give by

$$
\mathcal{L}(w, t, \alpha, \gamma) = d^T w + ct + y^T w + \frac{\rho}{2} \| w - z \|_2^2 - \gamma^T w + \alpha^T (Dw - t\mathbf{1})
\tag{5}
$$

We have introduced dual variables $\alpha$ and $\gamma$ to enforce the non-negativity and $Dw \leq t\mathbf{1}$ constraints, respectively. Taking derivatives w.r.t. $t$ yields

$$\frac{\delta \mathcal{L}}{\delta t} = c - \alpha^T \mathbf{1} \tag{6}$$

If $c - \alpha^T \mathbf{1} \neq 0$ then we can set $t$ such that the above is arbitrarily negative. Therefore, we assume that $c = \alpha^T 1$. This leads to the problem

$$\mathcal{L}(w, \alpha, \gamma) = d^T w + y^T w + \frac{\rho}{2}\|w - z\|_2^2 - \gamma^T w + \alpha^T Dw \quad \text{subject to} \quad c = \alpha^T 1 \tag{7}$$

Next, the derivative w.r.t. $w$ is

$$\frac{\delta \mathcal{L}}{\delta w} = d + y + \rho(w - z) - \gamma + D\alpha \tag{8}$$

which implies

$$w = \rho^{-1}(\rho z - d - y + \gamma - D\alpha) \tag{9}$$

Strong duality holds and the KKT conditions imply that $\gamma, \alpha \geq 0$ with $w^T \gamma = 0$ and $\alpha_i = 0$ if $D_i w_i < \|Dw\|_\infty$. Thus, unless $w = 0$, $\gamma^T \alpha = 0$. To check whether $w = 0$, we plug into (9) and check whether

$$0 = \rho z - d - y + \gamma - D\alpha \tag{10}$$

with $c = \alpha^T 1$. Let $q = (\rho z - d - y)_+$ where $(x)_+ = \max(0, x)$ applies element-wise. Since $\gamma$ can add arbitrarily positive amounts, (10) is equivalent to

$$0 = q - D\alpha \tag{11}$$

With the looser restriction $c \geq \alpha^T 1$. Thus, $D^{-1}q = \alpha$ and so

$$w = 0 \Leftrightarrow 1^T D^{-1} q \leq c \tag{12}$$

Next, assume that $w \neq 0$. Then to find $\alpha$, let $f = Dq$ and suppose that $f$ is sorted in *decreasing* order and that $w, D, \alpha$ are also sorted so that indices match up. This is always possible by permuting the vectors. Then $w_1$ is maximal iff

$$D_{11}w_1 = f_1 - D_{11}^2 c \geq f_2 = D_{22}w_2 \tag{13}$$

If $w_1$, $w_2$ are maximal, then

$$f_1 - D_{11}^2 \alpha_1 = f_2 - D_{22}^2(c - \alpha_1) \geq f_3 \tag{14}$$

Solving for $\alpha_1$ we find

$$\alpha_1 = \frac{f_1 - f2 + D_{22}^2 c}{D_{11}^2 + D_{22}^2} \tag{15}$$

And hence

$$f_1 - D_{11}^2 \frac{f_1 - f_2 + D_{22}^2 c}{D_{11}^2 + D_{22}^2} = \frac{D_{22}^2 f_1 + D_{11}^2 f_2 + D_{11}^2 D_{22}^2 c}{D_{11}^2 + D_{22}^2} \geq f_3 \tag{16}$$

Continuing on, suppose that $w_1$, $w_2$, $w_3$ are maximal so that

$$f_1 - D_{11}^2 \alpha_1 = f_2 - D_{22}^2 \alpha_2 = f_3 - D_{33}^2(c - \alpha_1 - \alpha_2) \geq f_4 \tag{17}$$

Solving that for $\alpha_1$ we find

$$\alpha_1 = \frac{f_1 - f_2 + D_{22}^2 \alpha_2}{D_{11}^2} \tag{18}$$

And then for $\alpha_2$

$$\alpha_2 = \frac{D_{11}^2(f_2 - f_3) - D_{33}^2(f_1 - f_2) + D_{11}^2 D_{33}^2 c}{D_{11}^2 D_{22}^2 + D_{11}^2 D_{33}^2 + D_{22}^2 D_{33}^2} \tag{19}$$

Thus, the maximal elements are given by

$$f_2 - D_{22}^2 \alpha_2 = \frac{D_{11}^{-2} f_1 + D_{22}^{-2} f_2 + D_{33}^{-2} f_3 - c}{D_{11}^{-2} + D_{22}^{-2} + D_{33}^{-2}} \tag{20}$$

It can be shown by induction that there are $k$ maximal elements only if

$$\frac{\sum_{i=1}^{k} D_{ii}^{-1} q_i - c}{\sum_{i=1}^{k} D_{ii}^{-2}} \geq D_{k+1,k+1} q_{k+1} \tag{21}$$

In order to recover $w$, we set $w_i = q_i$ if $D_{ii} w_i$ is not maximal, and if it is, we set $w_i = D_{ii}^{-1} \frac{\sum_{j=1}^{k} D_{jj}^{-1} q_j - c}{\sum_{j=1}^{k} D_{jj}^{-2}}$.

Next, we show that it is possible to quickly find $k$ in linear time (i.e. without sorting). Define $m(k) = \frac{\sum_{i=1}^{k} D_{ii}^{-2} f_i - c}{\sum_{i=1}^{k} D_{ii}^{-2}}$ and suppose that there are $k$ true maximal elements so that

$$m(k) > f_{k+1} \tag{22}$$

We show that $m(t) \geq f_{t+1}$, when $t > k$. Using the fact that $m(k) > f_{t+1}$,

$$m(k) > f_{t+1} \Leftrightarrow \sum_{i=1}^{k} D_{ii}^{-2} f_i - c > \left( \sum_{i=1}^{k} D_{ii}^{-2} \right) f_{t+1} \tag{23}$$

$$\Leftrightarrow \sum_{i=1}^{k} D_{ii}^{-2} f_i - c + \sum_{i=k+1}^{t} D_{ii}^{-2} f_i > \left( \sum_{i=1}^{k} D_{ii}^{-2} \right) f_{t+1} + \left( \sum_{i=k+1}^{t} D_{ii}^{-2} \right) f_{t+1} \tag{24}$$

$$\Leftrightarrow \sum_{i=1}^{t} D_{ii}^{-2} f_i - c > \left( \sum_{i=1}^{t} D_{ii}^{-2} \right) f_{t+1} \Leftrightarrow m(t) > f_{t+1} \tag{25}$$

Thus $m(t) \leq f_{t+1}$ for $t < k$ and $m(t) > f_{t+1}$ for $t \geq k$.

We can use this as a search criteria to develop an algorithm akin to the linear time median finding algorithm. This allows us to find $k$ in linear time without requiring that $f$ be sorted.

## 1.2 Solving for $z$

For $z$ the relevant parts are

$$\begin{aligned}\underset{z}{\text{minimize}} \quad & -y^T z + \frac{\rho}{2}\|w - z\|_2^2 \\ \text{subject to} \quad & Xz \geq \mathbf{1}\end{aligned} \tag{26}$$

This is easiest to solve by taking the dual. The Lagrangian is given by

$$\mathcal{L}(z, \alpha) = -y^T z + \frac{\rho}{2}\|w - z\|_2^2 + \alpha^T(\mathbf{1} - Xz) \tag{27}$$

Solving for $z$ we find

$$\frac{\delta \mathcal{L}}{\delta z} = -y - \rho w + \rho z - X^T \alpha = 0 \tag{28}$$

$$z = \rho^{-1}(y + \rho w + X^T \alpha) \tag{29}$$

Strong duality obtains, so plugging (29) into the Lagrangian yields the dual optimization problem

$$\begin{aligned}\underset{\alpha}{\text{minimize}} \quad & -(\rho 1 - X(y + \rho w))^T \alpha + \frac{1}{2}\alpha^T H \alpha \\ \text{subject to} \quad & \alpha \geq \mathbf{0}\end{aligned} \tag{30}$$

where $H = XX^T$.

## 2 Matrix Entries

This section explores the structure of $H = XX^T$. We assume that $N$ documents are compressed jointly, each of size $n_i$, and that pointers respect document boundaries. We show that $H$ is a $(k-1)$-banded matrix and that it is block diagonal with $N$ blocks, each of size $n_i \times n_i$ and corresponding to document $i$. This structure occurs when we assume a specific ordering for the set of potential pointers $\mathcal{P}$. In particular, pointers are ordered lexicographically according to the document they pertain to, then the length of their substring, and finally the location in which they insert their substring.

Recall that column $j$ of $X$ corresponds to pointer $p_j \in \mathcal{P}$ and that this column only has 1's at locations corresponding to words that $p_j$ can reconstruct. Let $m_i = \sum_{t=1}^{k} n_i - t + 1$ be the total number of pointers pertaining to document $i$. Since pointers respect document boundaries, our ordering implies that $X$ is a block diagonal matrix in which columns $1, \ldots, m_1$ can only have $1's$ in rows $1, \ldots, n_1$; columns $m_1 + 1, \ldots, m_1 + m_2$ can only have 1's in rows $n_1 + 1, \ldots, n_1 + n_2$; and so on. This immediately implies that $H$ is also a block diagonal matrix comprised of $N$ blocks, each of size $n_i \times n_i$ with the $i^{th}$ block corresponding to document $i$.

Next, to show that $H$ is $(k-1)$-banded, notice that each column of $X$ has a contiguous sequence of at most $k$ ones and is 0 everywhere else. The outer product $XX^T = \sum_{i=1}^{m} X_i X_i^T$ where $X_i$ is the $i^{th}$ column of $X$ is therefore formed by adding together a series of rank one matrices, each of which is $(k-1)$-banded. This implies that $H$ must itself be $(k-1)$-banded. '