[Reviews · NeurIPS 2013]

Submitted by Assigned_Reviewer_1

The method consists of compressing documents for accelerating the
document processing in other tasks, such as classification. The
coding scheme is related to the one used in the Lempel-Ziv algorithm,
storing pointers of substrings appearing at several locations in the
document. The proposed approach is formulated as a combinatorial
optimization problem, whose solution is approximated by a sequence
of convex problems solved by ADMM.
The experiments are carried on text classification problems, where
compressing the documents leads to some gains in memory and
computational efficiency, at a minor loss in terms of precision.

I found the approach interesting, even though I am not familiar enough
with the NLP literature to exactly judge the novelty of the approach.
I have only a few minor remarks to make
- the sentence ``an optimal lossless compression of D...'' requires some
clarifiation. Is the coding scheme optimal in terms of minimum entropy?
- it is not clear that the reweighting scheme can be interpreted here as
a majorization-minimization procedure. Is it really the case here?
- minimizing (4) with respect to w amounts to computing the Fenchel conjugate
of the (weighted) l_infty-norm, which involves a projection on a weighted
l1-norm (the dual norm of the l_infty-norm). When adding the non-negativity
constraint, this involves a projection on the simplex. Algorithms for
projecting on the simplex have been known for a while, and are similar to
the approach described in the paper. See
Brucker. An O(n) algorithm for quadratic knapsack problems. 1984.
see also
Bach et al. Optimization with Sparsity-Inducing Penalties. 2012,
for the computation of Fenchel conjugates for norms.
Summary: The paper proposes a way to compress documents to accelerate the document processing in various tasks. It seems that the approach is novel and performs well.

Submitted by Assigned_Reviewer_6

The authors proposed to compress the features by optimizing a formulation that takes into consideration both the pointer cost and the dictionary cost. The non-convex formulation is relaxed by replacing the binary constraints with box constraints in the interval of 0 and 1, and ADMM is applied for solving the relaxed problem. Experimental results were reported in comparison with several approaches.

The proposed compressive feature learning utilizes a losses compression formulation that considers both the pointer cost and the dictionary cost. The resulting problem is non-convex due to the binary constraints. The proposed formulation seems new.

In solving the relaxation, ADMM is adopted. The proposed approach requires solving a series of relaxed problems that are dependent on the parameter d. These series of solutions adopt a reweighting scheme. How about the efficiency of the proposed compression approach? For subsequent classification, it was reported that the classification speed can be improved with the proposed approach, which is reasonable.

After the features are extracted, the elastic net method is used for classification. The elastic net method has two tuning parameters, how are these two parameters tuned via cross validation? Two-dimensional grid search?

After reading other reviewers' comments and the author response, the reviewer would like to keep the original recommendation.
Summary: The authors proposed to compress the features by optimizing a formulation that takes into consideration both the pointer cost and the dictionary cost. The non-convex formulation is relaxed by replacing the binary constraints with box constraints in the interval of 0 and 1, and ADMM is applied for solving the relaxed problem.

Submitted by Assigned_Reviewer_8

This paper formulates the problem of selecting a subset of informative n-grams for document classfication as a lossless compression problem solved by iterative relaxation of he original hard combinatorial problem. While unsupervised feature learning as compression is not a new idea, this particular formulation is interesting, seems novel, and performs fairly well in small-scale experiments. The paper is very well written, the ideas are clear and reasonably motivated. The algorithm presentation, while not fully self-contained (there's a substantial supplement), is understandable as given. I would have liked more analysis of the algorithm's computational properties, or at least some experiments on computational cost-accuracy tradeoffs to help understand the scalability of the method (including the claims of parallelizability). Still in the experimental side, I'd also liked to see comparisons with popular lossy representation methods such as embeddings (Bengio et all 2006, Collobert and Weston 2008, Mikolov et al 2010, inter alia). And I'd like to see the tradeoffs between model size and accuracy obtained with this method compared with sparsifying regularization over the uncompressed n-gram features.
Summary: Formalizes n-gram feature selection for document classification as lossless compression, with an efficient relaxation algorithm for the original hard combinatorial problem. Elegant and well-motivated approach, with basic positive results but in some need of more analysis and experimentation.
Author Feedback

Author rebuttal: Thank you for reading our paper and providing helpful reviews and comments. We are very excited about our method because it addresses a fundamental issue when working with text – which n-grams to use for learning. Since the method is unsupervised, it can be run a single time offline to provide a succinct feature set for all subsequent learning tasks on the given data. When compared to a naive feature set containing all possible n-grams, the features our method selects can help elucidate structure in unsupervised problems and improve classification accuracy when there is a shortage of labeled data. Even when there are enough labels to handle the full n-gram feature set, our features achieved the same classification performance despite being two orders of magnitude fewer and having been selected without the use of labels. The reduced feature set also expedites experimentation as it allows training algorithms to run substantially faster.

We now focus on some of the questions and comments from our reviewers. Thank you for pointing out the confusing use of “optimal lossless compression”; we use it to mean optimal with respect to our objective and will reword it in our paper. Next, the reweighting scheme we employ can be interpreted as a Majorization-Minimization scheme very similar to the one in “Enhanced sparsity by reweighted l1 minimization” by Candes, Wakin, and Boyd, and we will include a proof of this in the appendix. In addition, thank you for pointing out the paper by Brucker. We were unaware of this algorithm but will cite it since, as you correctly point out, minimizing the augmented Lagrangian over the infinity norm and non-negativity constraint (i.e. w.r.t. w in (4)) can be converted into a quadratic knapsack problem by taking the dual.

As discussed in the paper, each pass of ADMM is linear in the document length and easily parallelizable. The algorithm we used in the paper was fast enough to run all of the datasets on a desktop computer and fully utilize its multiple cores. We agree that a table or graph demonstrating the algorithm’s scalability and parallelism would be useful, and while we omitted this for the sake of brevity, we can include it in the appendix. It is also worthwhile mentioning that the algorithm discussed in the paper is sufficiently parallelizable and vectorizable that we are implementing it on a GPU. Because of the NIPS page limit, we plan to write an extended version of the paper that more thoroughly explores our method’s performance in terms of accuracy in supervised tasks and in terms of computational efficiency. Finally, yes, all Elastic-Net parameters were chosen via cross-validated grid search.

Once again, we would like to thank our reviewers for their time and insightful comments.